# Results of a Randomized, Double-Blind, Placebo-Controlled, Phase 1b/2 Trial of Nabpaclitaxel + Gemcitabine ± Olaratumab in Treatment-Naïve Participants with Metastatic Pancreatic Cancer

**DOI:** 10.3390/cancers16071323

**Published:** 2024-03-28

**Authors:** Faithlore P. Gardner, Zev A. Wainberg, Christos Fountzilas, Nathan Bahary, Mark S. Womack, Teresa Macarulla, Ignacio Garrido-Laguna, Patrick M. Peterson, Erkut Borazanci, Melissa Johnson, Matteo Ceccarelli, Uwe Pelzer

**Affiliations:** 1Florida Cancer Specialists and Research Institute, Cape Coral, FL 34471, USA; 2UCLA School of Medicine, Los Angeles, CA 90095, USA; 3Roswell Park Comprehensive Cancer Center, Buffalo, NY 14203, USA; christos.fountzilas@roswellpark.org; 4Allegheny Health Network Cancer Institute, Pittsburgh, PA 15212, USA; nathan.bahary@ahn.org; 5Tennessee Oncology, Chattanooga, TN 37129, USA; 6Hospital Vall d’Hebrón, Vall d’Hebrón Institute of Oncology (VHIO), 08035 Barcelona, Spain; 7Department of Internal Medicine, Huntsman Cancer Institute at University of Utah, Salt Lake City, UT 84112, USA; 8Eli Lilly & Company, Indianapolis, IN 46285, USA; 9HonorHealth Research Institute, Scottsdale, AZ 85258, USA; 10Tennessee Oncology, Nashville, TN 37203, USA; 11Eli Lilly & Company, Sesto Fiorentino, 50019 Florence, Italy; 12Medical Department, Division of Hematology, Oncology and Tumorimmunology, Charité-Universitätsmedizin Berlin, Corporate Member of Freie Universität Berlin and Humboldt Universität zu Berlin, Charitéplatz 1, 10117 Berlin, Germany

**Keywords:** olaratumab, gemcitabine, nabpaclitaxel, metastatic pancreatic cancer

## Abstract

**Simple Summary:**

Nabpaclitaxel plus gemcitabine is the standard of care treatment for advanced pancreatic ductal adenocarcinoma (PDAC), the third leading cause of cancer-related deaths worldwide. This therapy reported improved overall survival in metastatic PDAC. Despite the efficacy of available treatments, the poor outcome for patients with PDAC necessitates the need for novel therapies. In a preclinical study, a neutralizing monoclonal antibody specific to human platelet-derived growth factor receptor alpha (PDGFRα) has proven its efficacy in advanced PDAC. Further, a PDGFRα monoclonal antibody, olaratumab, in combination with chemotherapy demonstrated clinical benefit in patients with soft tissue sarcoma. In light of the above findings, the current trial evaluates the efficacy and safety of olaratumab and nabpaclitaxel plus gemcitabine combination therapy in treatment-naïve patients with metastatic PDAC. However, the combination therapy failed to improve survival outcomes compared with chemotherapy alone. Nevertheless, the safety profile of olaratumab was consistent with the previous studies.

**Abstract:**

The efficacy and safety of olaratumab plus nabpaclitaxel and gemcitabine in treatment-naïve participants with metastatic pancreatic ductal adenocarcinoma was evaluated. An initial phase 1b dose-escalation trial was conducted to determine the olaratumab dose for the phase 2 trial, a randomized, double-blind, placebo-controlled trial to compare overall survival (OS) in the olaratumab arm vs. placebo arms. In phase 1b, 22 participants received olaratumab at doses of 15 and 20 mg/kg with a fixed dose of nabpaclitaxel and gemcitabine. In phase 2, 159 participants were randomized to receive olaratumab 20 mg/kg in cycle 1 followed by 15 mg/kg in the subsequent cycles (n = 81) or the placebo (n = 78) on days 1, 8, and 15 of a 28-day cycle, plus nabpaclitaxel and gemcitabine. The primary objective of the trial was not met, with a median OS of 9.1 vs. 10.8 months (hazard ratio [HR] = 1.05; 95% confidence interval [CI]: 0.728, 1.527; *p* = 0.79) and the median progression-free survival (PFS) was 5.5 vs. 6.4 months (HR = 1.19; 95% CI: 0.806, 1.764; *p* = 0.38), in the olaratumab vs. placebo arms, respectively. The most common treatment-emergent adverse event of any grade across both arms was fatigue. Olaratumab plus chemotherapy failed to improve the OS or PFS in participants with metastatic PDAC. There were no new safety signals.

## 1. Introduction

Pancreatic ductal adenocarcinoma (PDAC) is the third leading cause of cancer-related deaths worldwide. In the United States, the mortality rate was estimated to be 4.1 (per 100,000) in women and 5.7 in men (per 100,000) [1].

Nabpaclitaxel plus gemcitabine is an established first-line standard care treatment option for patients with advanced disease [2,3]. First-line standard care treatment has not changed over the years [3]. In the MPACT phase 3 clinical trial, nabpaclitaxel plus gemcitabine demonstrated an improved overall survival (OS), progression-free survival (PFS), and response rate, in comparison to gemcitabine alone in participants with metastatic PDAC, but with increased rates of peripheral neuropathy and myelosuppression [4]. 

Despite the availability of treatments, the overall prognosis for patients with PDAC remains extremely poor, indicating an unmet need for novel therapies [5]. Platelet-derived growth factor receptor alpha (PDGFRα) is overexpressed in PDAC and contributes to the activation of pancreatic stellate cells and the formation of the characteristic desmoplastic stroma [6,7]. Preclinical studies in tumor xenografts have indicated that a neutralizing monoclonal antibody specific to human PDGFRα could be a potential therapeutic target in advanced (unresectable locally advanced or metastatic) PDAC [8,9]. Moreover, a trial in participants with soft tissue sarcoma (STS) wherein platelet-derived growth factor (PDGF)/PDGF receptor (PDGFR) signaling plays a significant role, reported a clinical benefit in participants treated with olaratumab (PDGFRα monoclonal antibody) [10] in combination with chemotherapy [11].

Given the significant improvement in overall survival in STS, and the similar tumor microenvironment of PDAC, a trial was conducted to evaluate the efficacy and safety of olaratumab with standard first-line nabpaclitaxel plus gemcitabine in treatment-naïve participants with metastatic PDAC.

## 2. Materials and Methods

### 2.1. Trial Design, Participants, and Intervention

The trial (NCT03086369), which encompassed both phase 1b and phase 2 components, aimed to assess the effectiveness and safety of olaratumab in combination with nabpaclitaxel and gemcitabine, as compared to a placebo combined with the same chemotherapy drugs. The study focused on participants diagnosed with metastatic PDAC (Figure 1). During the phase 1b segment, an open-label, dose-finding approach was employed. In contrast, the phase 2 portion followed a multicenter, randomized, double-blind, parallel, and placebo-controlled design. Eligibility criteria included participants aged 18 years or older, with a histological or cytological diagnosis of adenocarcinoma of the exocrine pancreas at a metastatic stage (according to AJCC v8). Additionally, participants needed to have measurable disease as defined by Response Evaluation Criteria in Solid Tumors (RECIST) Version 1.1 and adequate organ function, along with an Eastern Cooperative Oncology Group performance status of 1 or better. Participants with endocrine pancreatic tumors or ampullary cancer, with central nervous system metastasis, or who had received prior treatment with nabpaclitaxel in adjuvant or neo-adjuvant settings were excluded from the trial.

Phase 1b was an open-label, single-arm, 3 + 3 dose-escalation trial that evaluated the safety and tolerability of olaratumab plus nabpaclitaxel and gemcitabine to determine the maximum tolerated dose (MTD) of olaratumab. This phase consisted of the following cohorts: in Cohort 1, the starting dose of olaratumab of 15 mg/kg was administered on days 1, 8, and 15 of a 28-day cycle, with the plan to potentially escalate the dose to 20 mg/kg on days 1, 8, and 15 in Cohort 2 or to 20 mg/kg on Days 1 and 15 in Cohort 3, followed by 25 mg/kg on days 1 and 15 in Cohort 4. In addition, a cohort expansion was added to ensure an appropriate number of participants were evaluated for safety at the highest tolerated dose before opening the phase 2 trial. During the dose escalation, continuous evaluation of toxicity in dose cohorts was performed. In the context of this trial, dose-limiting toxicity (DLT) refers to adverse events occurring during cycle 1 that are deemed related to the trial drugs by the investigator. The MTD of olaratumab was determined as the highest dose level at which no more than 33% of participants experienced a DLT during cycle 1. If no DLT occurred in a cohort of 3 participants, a new cohort of 3 participants received treatment at the next higher dose level. However, if 1 out of 3 participants at any dose level encountered a DLT, that cohort was expanded to 6 participants. Further dose escalation was considered if fewer than 2 out of 6 evaluable participants experienced a DLT. DLTs included events such as febrile neutropenia, grade 4 thrombocytopenia, or grade 3 thrombocytopenia with clinically significant hemorrhage, grade 4 neutropenia lasting 7 days or longer, non-hematologic grade ≥3 toxicity (with some exceptions), or any other toxicity deemed significant by the primary investigator (details provided in the Appendix A).

In phase 2, participants were randomly assigned 1:1 to receive olaratumab (at the phase 1b dose and schedule) or placebo, followed by 125 mg/m^2^ nabpaclitaxel and 1000 mg/m^2^ gemcitabine infusion. The randomization was stratified into 4 groups, one for each combination of the two baseline factors: age group (<70 years vs. ≥70 years) and prior adjuvant/neo-adjuvant gemcitabine use (yes vs. no). Participants received the study treatment until disease progression or until any criterion for discontinuation was met.

### 2.2. Outcomes

The primary objective of the phase 1b trial was to determine the recommended phase 2 dose (RP2D) of olaratumab in combination with nabpaclitaxel plus gemcitabine. The RP2D was determined based on the MTD, DLTs, and other safety endpoints including treatment-emergent adverse events (TEAEs) and serious adverse events (SAEs). 

The primary objective of the phase 2 trial was to determine the OS efficacy of olaratumab in combination with nabpaclitaxel plus gemcitabine as compared to gemcitabine alone. Secondary objectives included the evaluation of other efficacy parameters such as the PFS, objective response rate (ORR), and duration of response (DoR); safety and tolerability; patient-reported outcomes (PROs); pharmacokinetics (PK); and immunogenicity. The evaluation of PDGFR α and β expression as a potential biomarker was an exploratory objective.

### 2.3. Sample Size Determination and Statistical Analysis 

The phase 1b part of the trial was designed to enroll up to 24 participants using a 3 + 3 dose-escalation design, with additional participants added to a cohort expansion at the highest tolerated dose. For the phase 2 part, assuming a true OS hazard ratio (HR) of 0.67 and an 80% statistical power to detect a significant difference in the OS between the trial arms, approximately 162 participants were needed to be enrolled and randomly assigned 1:1 to olaratumab and placebo treatments.

The statistical analyses for this trial were both descriptive and inferential. Unless otherwise specified, treatment effects were evaluated based on a two-sided significance level of 0.05. For continuous variables, summary statistics included the number of participants, mean, median, standard deviation, minimum, and maximum. Categorical endpoints were summarized using the number of participants, frequency, and percentages. The OS was defined as the time from the date of randomization to the date of death from any cause. The PFS was defined as the time from randomization until the first radiographic documentation of progression as defined by RECIST (Version 1.1) or death from any cause in the absence of progressive disease. The OS and PFS were compared between treatment arms using the stratified logrank test, and the corresponding HR between treatment arms was estimated using a stratified Cox regression model (Cox 1972), stratified by the randomization strata. For the OS and PFS, the median survival time and survival rates at various time points with a 95% confidence interval (CI) for each treatment arm were estimated using Kaplan–Meier (Kaplan and Meier 1958) summary statistics. The ORR, with 95% CI, was summarized for each treatment arm and compared between treatment arms using the Cochran–Mantel–Haenszel test, adjusting for the randomization strata. All participants who received at least 1 dose of any study therapy were evaluated for safety and toxicity. Adverse events and their severity were classified using the Common Terminology Criteria for Adverse Events Version 4.0 and, if necessary, the Medical Dictionary for Regulatory Activities Version 23.1.

### 2.4. Pharmacokinetics and Immunogenicity

PK analyses were conducted for participants who received at least 1 dose of the study drug (olaratumab) and had samples collected from phase 1b and phase 2 parts of this trial. 

Population PK parameters were estimated using NONMEM (Version 7.4.2, ICON Development Solutions, Ellicott City, MA, USA) by combining phase 1b and phase 2 data with a previous analysis dataset using a population PK model developed from all previous data. Immunogenicity samples underwent analysis to detect anti-olaratumab antibodies at Pharmaceutical Product Development, LLC in Richmond, VA, USA. This assessment utilized a validated immunoassay and followed a 4-tiered approach. The process involved detecting, confirming, and determining titers and characterizing the neutralizing activity of these antibodies in human serum. To optimize the anti-olaratumab antibody assay, specific conditions were met: a minimal dilution of 1:10, a validated sensitivity of 13.7 ng/mL, and a drug tolerance exceeding 500 µg/mL olaratumab in the presence of 500 ng/mL affinity-purified hyper-immunized monkey anti-olaratumab antibody.

### 2.5. Patient-Reported Outcomes

The modified Brief Pain Inventory-short form (mBPI-sf) and the European Organisation for Research and Treatment of Cancer Quality of Life survey (EORTC QLQ-C30) were completed on day 1 of each cycle of treatment throughout the trial. Paper versions of each questionnaire were used in this trial. The primary efficacy assessment for pain was based on the time to the first worsening of the mBPI-sf “worst pain” score. In this context, “worsening” was defined as either an increase of more than 2 points from the baseline “worst pain” score or an escalation in the analgesic drug class by more than 1 level. The analysis of the time to first pain worsening utilized the Kaplan–Meier method, and a comparison was performed between the two treatment arms using a logrank test.

## 3. Results

### 3.1. Participant Disposition and Baseline Characteristics

In phase 1b, a total of 22 participants were treated (cohort 1: olaratumab 15 mg/kg plus nabpaclitaxel and gemcitabine [n = 3] on days 1, 8, and 15 of a 28-day cycle; cohort 2: olaratumab 20 mg/kg plus nabpaclitaxel and gemcitabine [n = 7] on days 1, 8, and 15; cohort expansion: olaratumab 20 mg/kg plus nabpaclitaxel and gemcitabine [n = 12] on days 1, 8, and 15). Demographics were generally well balanced across three cohorts; the mean age of participants enrolled was 67.7 years, and 14 participants (63.6%) were male. The mean body-mass index was higher in cohort 1 (29.9 kg/m^2^) than cohort 2 (27.7 kg/m^2^) and the expansion cohort (27.2 kg/m^2^). Overall, no substantial differences in terms of baseline demographics and clinical characteristics were noted across the three cohorts. The data are summarized in Appendix A.

In phase 2, 159 participants were randomly assigned to the olaratumab arm (n = 81) and the placebo arm (n = 78), and at the time of trial closure, of these 159 patients, 156 participants (98.14%) discontinued the study treatment: the olaratumab arm (n = 80, 98.8%) and the placebo arm (n = 76, 97.4%). The major reason for discontinuation was disease progression (Figure 2).

A total of 95 participants (59.7%) were male, with a mean age of 65.5 years, and most of them (82.0%) were <75 years. In phase 2, the demographic and baseline characteristics were generally balanced between both arms. A total of 22 (27.2%) participants in the olaratumab arm and 17 (21.8%) participants in the placebo arm underwent a prior surgical procedure either with curative or palliative intent (Table 1).

### 3.2. Phase 1b DLTs and Evaluation of MTD

Both the doses of olaratumab in combination with nabpaclitaxel and gemcitabine were safe and well tolerated in participants with PDAC, with AEs consistent in nature and within the frequency range expected for chemotherapy with nabpaclitaxel plus gemcitabine or olaratumab therapy. There was 1 DLT reported with the 20 mg/kg dose (grade 4 neutropenia that lasted for >7 days). Although the MTD for olaratumab in combination with nabpaclitaxel and gemcitabine was not reached, safety and exposure–response analyses from various olaratumab studies led to the establishment of the RP2D. For this combination, the RP2D involves an initial loading dose of 20 mg/kg of olaratumab administered on days 1, 8, and 15 during cycle 1, followed by a maintenance dose of 15 mg/kg on the same days during subsequent cycles (cycle 2 and beyond).

### 3.3. Efficacy of Phase 1b

Of the 22 participants, 5 (22.7%) were discontinued prior to the first assessment, and 16 (72.7%) were evaluable for tumor response. Six (37.5%) partial responses were observed, and eight (50.0%) had stable disease. The ORR was 20.7%, and the disease control rate was 48.3%.

### 3.4. Efficacy of Phase 2

The median OS was 9.1 months in the olaratumab arm and 10.8 months in the placebo arm (Figure 3); the difference was not statistically significant (HR = 1.05; 95% CI: 0.728, 1.527; *p* = 0.79). The primary objective of the trial was not met.

The median PFS was 5.5 months in the olaratumab treatment arm and 6.4 months in the placebo arm (Figure 4). This difference was not statistically significant (HR = 1.19; 95% CI: 0.806, 1.764; *p* = 0.377).

Overall response rates and disease control rates were comparable for both the olaratumab (30.5%, 69.5%) and placebo arms (33.8%, 77.5%), respectively (Table 2). The median DoR was 5.6 months in both arms (HR = 1.23; 95% CI: 0.61, 2.47; *p* = 0.57).

### 3.5. Safety

In phase 1b, most participants reported at least 1 TEAE (n = 21, 95.5%). A total of 15 (68.2%) participants reported a TEAE of grade ≥3 severity (Table 3). In total, 9 participants (40.9%) reported at least ≥1 SAE (Appendix A). There were four (18.2%) deaths reported, of which one (4.5%) death occurred each during therapy and within 30 days of treatment discontinuation because of an AE of a confusional state (n = 1, 4.5%) and study disease (n = 1, 4.5%); and two (9.1%) deaths occurred after 30 days of treatment discontinuation because of study disease. The AE of the confusional state was assessed as related to gemcitabine by the investigator.

In phase 2, a total of 80 participants (98.8%) in the olaratumab arm and 78 participants (100.0%) in the placebo arm reported ≥1 TEAE. Amongst these, 71 participants (87.7%) had TEAEs of grade ≥3 severity in the olaratumab arm and 72 participants (92.3%) in the placebo arm. The most common TEAE across both the arms of any grade was fatigue and of grade ≥3 was a decreased neutrophil count (Table 4). 

The most common TEAEs related to the study treatment as assessed by the investigator are presented in Table 5. 

In phase 2, 91 participants (57.2%) reported ≥1 SAE (Appendix A). A total of 114 (71.7%) deaths occurred in the trial, of which 56 (69.1%) deaths were in the olaratumab arm and 58 (74.4%) deaths were in the placebo arm. In the olaratumab arm, 20 (24.7%) deaths occurred during therapy or within 30 days of treatment discontinuation, and 36 (44.4%) deaths occurred after 30 days of treatment discontinuation. The major reason for deaths during therapy or within 30 days of treatment discontinuation was study disease (n = 13, 16.0%) and AEs (n = 7, 8.6%). Of these seven AEs, respiratory failure (n = 1, 1.2%) was assessed as related to gemcitabine by the investigator.

The dose adjustments were balanced between the treatment and placebo arms. The majority of dose and drug administration modifications were completed due to the occurrence of AEs. Participants with at least one dose adjustment (reported in ≥4 participants) for olaratumab were 73 (90.1%) and 69 (88.5%), for nabpaclitaxel were 74 (91.4%) and 75 (96.2%), for gemcitabine were 73 (90.1%) and 74 (94.9%) in the olaratumab and placebo arms, respectively. 

### 3.6. Participant-Reported Outcomes

The mBPI-sf scores were recorded at the baseline. The mean (SD) worst pain in the last 24 h scores were 3.9 (2.6) and 4.2 (2.8), respectively. There were no statistically significant differences between treatment arms for the time to the first worsening of pain (median 14.1 for Arm A vs. 6.1 months for Arm B, HR = 0.496, 95% CI: 0.238-1.032, *p* = 0.056).

At the baseline, 70 participants in Arm A and 66 participants in Arm B completed the baseline Quality of Life Questionnaire (QLQ-C30). Mean (SD) baseline scores were 61.4 (23.0) and 61.1 (22.8) for global health status/quality of life and 76.7 (22.1) and 78.4 (18.5) for the physical functioning of participants enrolled in Arm A and Arm B, respectively. There were no differences between treatment arms for the time to first worsening on these subscales of the QLQ-C30.

### 3.7. Pharmacokinetics and Immunogenicity

The PK results of olaratumab from the phases 1b and 2 population PK analysis (phases 1b and 2 combined with previous studies) confirm that olaratumab serum concentrations were in the expected range as indicated by the similarities between the results of the current population PK analysis and that of previous analyses (Appendix A).

There were 81 evaluable participants for the antidrug antibody analysis (ADA). Among the evaluable participants, overall, 15 (18.5%) qualified for the ADA-positive response, and no participants qualified for the TE ADA-positive response. Since there were no TE ADA-positive participants, an assessment of the impact on olaratumab PK (exposure) was not performed.

## 4. Discussion

Despite available options for the treatment of metastatic pancreatic cancer, the overall prognosis for stage IV patients remains poor with a 5-year survival rate that is ≤3% [12,13,14].

PDGFR/PDGF signaling has been shown to create autocrine and paracrine signaling in pancreatic cancer, contributing to a stromal response and the activation of pancreatic stellate cells—features of pancreatic cancer and contributors to the pancreatic cancer microenvironment [15,16,17]. This trial was designed to evaluate whether olaratumab, an anti-PDGFRα recombinant human immunoglobulin G subclass 1–type monoclonal antibody in combination with nabpaclitaxel plus gemcitabine might improve survival as compared with chemotherapy alone.

The results from this trial showed that the addition of olaratumab to nabpaclitaxel and gemcitabine resulted in a median OS of 9.10 months in the olaratumab arm and 10.81 months in the placebo arm, a difference that was not statistically significant (HR = 1.054; 95% CI: 0.728, 1.527; *p* = 0.79) and, therefore, the trial did not meet its primary objective. Additionally, the combination did not demonstrate a statistically significant difference in the median PFS (5.5 months vs. 6.4 months; HR = 1.19; 95% CI: 0.806, 1.764; 95%; *p* = 0.38) or median DoR (5.6 months for both treatment arms; HR = 1.29; 95% CI: 0.61, 2.47; *p* = 0.57) compared to placebo. There was no statistically significant difference in the ORR (30.5% vs. 33.8%) and the disease control rate (69.5% vs. 77.5%) between the olaratumab and placebo arms. It should be noted that the OS observed in the placebo arm of this trial appears slightly improved compared to the OS experience by participants enrolled in the nabpaclitaxel plus gemcitabine arm of the landmark MPACT trial (median OS of 8.7 months) [18]; the median OS observed in the placebo arm of our trial is more in line with recent trials using nabpaclitaxel plus gemcitabine as the chemotherapy backbone. Several agents including ibrutinib [19] and necuparanib [20], tarextumab [21], apatorsen [22], cisplatin [23], enzalutamide [24], and momelotinib [25], have been studied in combination with nabpaclitaxel plus gemcitabine in PDAC, but only a few trials showed some potential preliminary benefits [26], suggesting the complexity of advancing new therapies in such a highly heterogenic disease. For example, the RESOLVE trial reported a median OS of 9.7 months with ibrutinib plus nabpaclitaxel plus gemcitabine and 10.8 months with placebo [19]. Another phase 2 trial comparing the standard nabpaclitaxel plus gemcitabine with or without necuparanib reported an OS of 10.7 months compared to 10 months in the placebo arm [20,26]. The improved OS observed in those studies and our trial could be reflective of the availability of second-line therapy options as well as better supportive care since nabpaclitaxel plus gemcitabine was established as a standard care option for the initial treatment of patients with metastatic PDAC [27]. Additionally, in our trial, a slightly higher number of participants in the olaratumab arm had received prior systemic anticancer therapy in the adjuvant or neo-adjuvant setting including gemcitabine-based regimens (23.5% in the olaratumab arm vs. 17.9% in the placebo arm); whether this difference contributed to the negative results of the trial is unclear. Finally, it should be noted that the trial was designed to allow a more inclusive enrollment of participants. This led to a trial population more reflective of a real-world PDAC patient population.

The recommended dosage for the randomized portion of this trial was established based on safety assessments and exposure–response analyses from various olaratumab trials. Participants received a 20 mg/kg olaratumab loading dose on days 1, 8, and 15 during cycle 1, followed by a 15 mg/kg maintenance dose on the same days in subsequent cycles. This dosing regimen was chosen to optimize the achievement of olaratumab serum concentrations associated with clinical efficacy while minimizing any reductions or omissions of the standard-of-care chemotherapy backbone. While, in phase 1b, the MTD of olaratumab was not reached, there was a trend toward a higher number of early grades of ≥3 neutropenia and a dose reduction in nabpaclitaxel plus gemcitabine with a 20 mg/kg dose relative to the 15 mg/kg olaratumab dose. In the phase 2 trial, the safety profile of olaratumab in combination with nabpaclitaxel plus gemcitabine showed anticipated toxicities, and no new safety issues were reported. Furthermore, there were no relevant differences in AEs potentially related to study treatment between the two arms; also, the rate and type of AEs and discontinuation due to AEs were comparable between olaratumab and the placebo.

Infusion-related reactions (IRRs) are a known Adverse Drug Reaction for olaratumab, and the risk of an anaphylactic reaction is associated with elevated IgE antibody levels [28]. As previously reported [28], olaratumab has a glycosylation site in the Fab region occupied by oligosaccharides capped with galactose-1,3-galactose (alpha-Gal) and/or N-glycolylneuraminic acid residues [29]. The hypothesis was that severe hypersensitivity reactions occurring during the initial infusion of olaratumab could be mediated by pre-existing IgE on mast cells and that IgE binding to the alpha-gal epitope in the Fab portion of the heavy chain of olaratumab could lead to the activation and degranulation of mast cells. A similar phenomenon was also observed and described with cetuximab [30]. In order to minimize the observed grade ≥3 IRRs in participants treated with olaratumab, the protocol was amended during the course of the trial to include testing for immunoglobulin E (IgE) anti-alpha-gal antibodies as part of the screening and exclude participants with antibody levels greater than the upper limit of normal (ULN). Prior to the protocol amendment, participants in this trial were tested for pre-existing IgE antibodies against α-gal as a preplanned retrospective analysis. However, the results of these tests were not available prior to the first dosing of olaratumab/placebo. In total, 61 participants were enrolled under this protocol amendment, and none of them had a grade ≥3 IRR. Overall, in the olaratumab arm, five (6.2%) participants had anti-alpha-gal IgE >ULN and one of those participants had grade ≥3 IRR. In the placebo arm, two (2.6%) participants had anti-alpha-gal IgE >ULN, and none of them had grade ≥3 IRRs.

The PK (exposure) for olaratumab in the phase 1b and phase 2 parts of the trial was similar to previous clinical trials [10] and, therefore, unlikely to have a significant impact on safety and efficacy outcomes. There were no significant differences between treatment arms in the time to the worsening of any PRO measures in this trial.

The exploratory analyses to correlate the expression of PDGFRα with clinical outcome were not conducted due to the overall negative efficacy results of the trial.

## 5. Conclusions

Olaratumab combined with nabpaclitaxel plus gemcitabine failed to improve the OS or PFS as compared to chemotherapy alone in participants with metastatic PDAC. The safety profile of olaratumab was consistent with the previous trials.

## Figures and Tables

**Figure 1 cancers-16-01323-f001:**
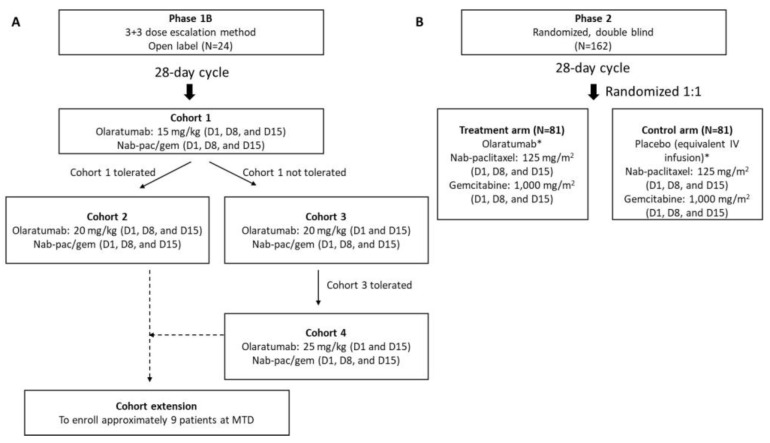
Trial design: (**A**) Phase 1b. (**B**) Phase 2. Phase 1b consisted of a 3 + 3 dose-escalation trial. The maximum tolerated dose was determined to be the highest olaratumab dose level at which no more than 33% of participants experienced a DLT during cycle 1. After the determination of the MTD, a Cohort expansion further evaluated the dose in additional participants. Abbreviations: D, day; DLT, dose-limiting toxicity; Gem, gemcitabine; IV, intravenous; MTD, maximum tolerated dose; N, number of participants; Nabpac, nabpaclitaxel. * The dose and schedule of olaratumab in phase 2 were determined based upon tolerability in the phase 1b part of the trial. Participants continued treatment until disease progression, unacceptable toxicities, or until other criteria for discontinuation were met.

**Figure 2 cancers-16-01323-f002:**
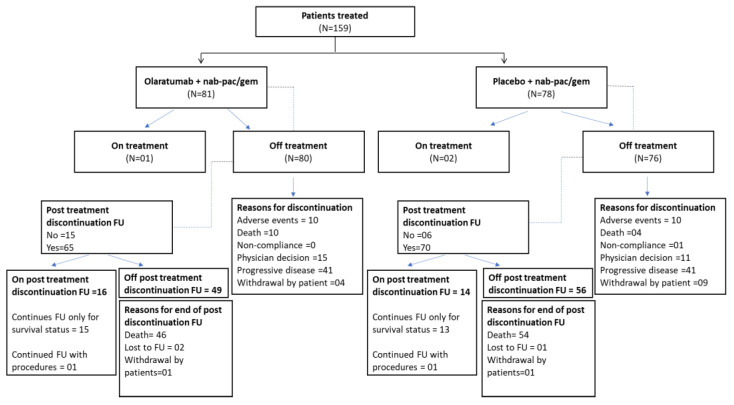
Participant disposition. Abbreviations: FU, follow up; Gem, gemcitabine; N, number of participants in safety population; n, number of participants within category; Nab-Pac, nabpaclitaxel. Data cut-off date: 30 July 2018 (Phase 1b); 5 January 2021 (Phase 2).

**Figure 3 cancers-16-01323-f003:**
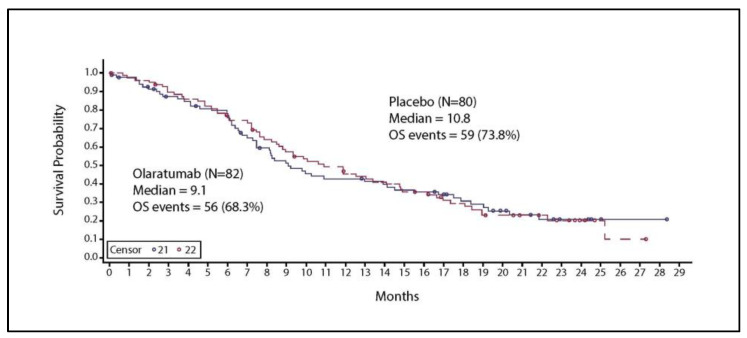
Kaplan–Meier curve for overall survival. Solid line represents patients with chemotherapy and Olaratumab treatment Abbreviations: N, total number of participants; OS, overall survival.

**Figure 4 cancers-16-01323-f004:**
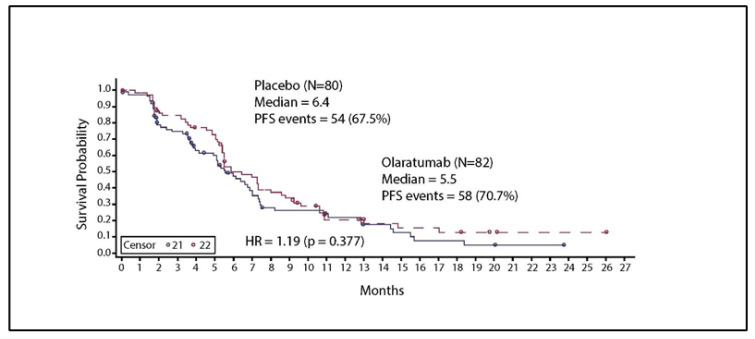
Kaplan–Meier curve for progression-free survival. Solid line represents patients with chemotherapy and Olaratumab treatment. Abbreviations: HR, hazard ratio; N, total number of participants; PFS, progression-free survival HR—Stratified. Stratified by interactive voice response system reported age group and prior adjuvant/neo-adjuvant gemcitabine use (yes vs. no).

**Table 1 cancers-16-01323-t001:** Phase 2: demographic and baseline characteristics.

Parameter	Olaratumab Arm (N = 81)	Placebo Arm (N = 78)	Total (N = 159)
Age, years	65.6 ± 9.4	65.5 ± 9.2	65.5 ± 9.3
Male	52 (64.2)	43 (55.1)	95 (59.7)
Race, White	71 (87.7)	71 (91.0)	142 (89.3)
BMI (kg/m^2^)	26.5 ± 5.5	26.2 ± 5.6	26.3 ± 5.5
Country, USA	75 (92.6)	71 (91.0)	146 (91.8)
Primary tumor present, yes	69 (85.2)	70 (89.7)	-
ECOG performance status			
0	34 (42.0)	38 (48.7)	-
1	46 (56.8)	40 (51.3)	-
2	1 (1.2)	0	-
Sites of metastatic disease ^1^			
Liver	44 (54.3)	45 (57.7)	-
Lung	16 (19.8)	12 (15.4)	-
Pancreas	25 (30.9)	33 (42.3)	-
Peritoneum	6 (7.4)	5 (6.4)	-
Lymph Node	6 (7.4)	9 (11.5)	-
Prior systemic therapy	19 (23.5)	14 (17.9)	33 (20.8)
Fluorouracil	13 (16.0)	7 (9.0)	20 (12.6)
Folinic Acid	12 (14.8)	5 (6.4)	17 (10.7)
Irinotecan	12 (14.8)	6 (7.7)	18 (11.3)
Gemcitabine	11 (13.6)	9 (11.5)	20 (12.6)
Oxaliplatin	11 (13.6)	6 (7.7)	17 (10.7)
Capecitabine	10 (12.3)	3 (3.8)	13 (8.2)
Cetuximab	0 (0.0)	1 (1.3)	1 (0.6)
Prior Surgical procedures			
Curative Surgery			
Pancreaticoduodenectomy	5 (6.2)	4 (5.1)	9 (5.7)
Pancreatectomy	4 (4.9)	2 (2.6)	6 (3.8)
Pancreatic Resection	1 (1.2)	2 (2.6)	3 (1.9)
Palliative Surgery			
Pancreaticoduodenectomy	8 (9.9)	6 (7.7)	14 (8.8)
Pancreatectomy	0	2 (2.6)	2 (1.3)
Pancreaticosplenectomy	1 (1.2)	1 (1.3)	2 (1.3)
Pancreatic Resection	1 (1.2)	0	1 (0.6)
Whipple	1 (1.2)	0	1 (0.6)
Whipple procedure	1 (1.2)	0	1 (0.6)

^1^ Participants may fall into more than 1 category. Abbreviations: BMI, body mass index; ECOG, Eastern Cooperative Oncology Group; N, number of participants in safety population; n, number of participants within a subcategory; SD, standard deviation. Data are mean ± SD or n (%). Data cut-off date = 5 January 2021.

**Table 2 cancers-16-01323-t002:** Best overall response.

Parameter	Olaratumab Arm		Placebo Arm	
	n (%)	95% CI	n (%)	95% CI
Best overall response				
Complete response	2 (2.4)	(0.3, 8.5)	2 (2.5)	(0.3, 8.7)
Partial response	23 (28.0)	(18.7, 39.1)	25 (31.3)	(21.3, 42.6)
Stable disease	32 (39.0)	(28.4, 50.4)	35 (43.8)	(32.7, 55.3)
Progressive disease	11 (13.4)	(6.9, 22.7)	7 (8.8)	(3.6, 17.2)
Non-evaluable	14 (17.1)	(9.7, 27.0)	11 (13.8)	(7.1, 23.3)
Overall response rate	25 (30.5)	(20.8, 41.6)	27 (33.8)	(23.6, 45.2)
Disease control rate	57 (69.5)	(58.4, 79.2)	62 (77.5)	(66.8, 86.1)

Abbreviations: CI, confidence interval; N, number of participants in safety population; n, number of participants within a subcategory; RECIST, Response Evaluation Criteria in Solid Tumors. The response criteria used was RECIST 1.1. Data cut-off date = 5 January 2021.

**Table 3 cancers-16-01323-t003:** Phase 1b: treatment-emergent adverse events occurring in ≥20% of drug-treated participants.

Preferred Term	Cohort 1 Olaratumab 15 mg/kg Plus NabPac Gem (N = 3)		Cohort 2 Olaratumab 20 mg/kg Plus NabPac Gem (N = 7)		Cohort Expansion Olaratumab 20 mg/kg Plus NabPac Gem (N = 12)		Total (N = 22)	
	Any Grade n (%)	Grade 3/4/5 n (%)	Any Grade n (%)	Grade 3/4/5 n (%)	Any Grade n (%)	Grade 3/4/5 n (%)	Any Grade n (%)	Grade 3/4/5 n (%)
Participants with ≥1 TEAE	3 (100.0)	3 (100.0)	7 (100.0)	6 (85.7)	11 (91.7)	6 (50.0)	21 (95.5)	15 (68.2)
Fatigue	2 (66.7)	0 (0)	3 (42.9)	0 (0)	6 (50.0)	0 (0)	11 (50.0)	0 (0)
Nausea	1 (33.3)	0 (0)	3 (42.9)	0 (0)	6 (50.0)	0 (0)	10 (45.5)	0 (0)
Constipation	2 (66.7)	0 (0)	3 (42.9)	0 (0)	2 (16.7)	0 (0)	7 (31.8)	0 (0)
Thrombocytopenia	2 (66.7)	0 (0)	1 (14.3)	1 (14.3)	4 (33.3)	1 (8.3)	7 (31.8)	3 (13.6)
Neutrophil count decreased	1 (33.3)	1 (33.3)	3 (42.9)	1 (14.3)	3 (25.0)	1 (8.3)	7 (31.8)	3 (13.6)
Diarrhea	2 (66.7)	0 (0)	1 (14.3)	0 (0)	3 (25.0)	0 (0)	6 (27.3)	0 (0)
Anemia	2 (66.7)	1 (33.3)	2 (28.6)	1 (14.3)	1 (8.3)	0 (0)	5 (22.7)	2 (9.1)
Infusion-related reactions	0 (0)	0 (0)	3 (42.9)	2 (28.6)	2 (16.7)	1 (8.3)	5 (22.7)	3 (13.6)

Abbreviations: Gem, gemcitabine; N, number of participants in Safety Population; n, number of participants within the category; NabPac = nabpaclitaxel; TEAE, treatment-emergent adverse event.

**Table 4 cancers-16-01323-t004:** Phase 2: treatment-emergent adverse events occurring in ≥25% of drug-treated participants.

Parameter	Olaratumab Arm		Placebo Arm	
	Any Grade n (%)	Grade 3/4/5 n (%)	Any Grade n (%)	Grade 3/4/5 n (%)
Fatigue	54 (66.7)	7 (8.6)	44 (56.4)	8 (10.3)
Anemia	46 (56.8)	19 (23.5)	45 (57.7)	20 (25.6)
Diarrhea	40 (49.4)	6 (7.4)	30 (38.5)	5 (6.4)
Nausea	33 (40.7)	2 (2.5)	39 (50.0)	2 (2.6)
Neutrophil count decreased	31 (38.3)	21 (25.9)	25 (32.1)	23 (29.5)
Platelet count decreased	29 (35.8)	6 (7.4)	31 (39.7)	11 (14.1)
Alopecia	28 (34.6)	0 (0)	16 (20.5)	0 (0)
Edema peripheral	25 (30.9)	0 (0)	28 (35.9)	2 (2.6)
Pyrexia	24 (29.6)	0 (0)	21 (26.9)	1 (1.3)
Constipation	22 (27.2)	0 (0)	30 (38.5)	0 (0)
Decreased appetite	22 (27.2)	1 (1.2)	24 (38.5)	0 (0)
Neutropenia	20 (24.7)	14 (17.3)	16 (20.5)	15 (19.2)
Dizziness	20 (24.7)	1 (1.2)	21 (26.9)	0 (0)
Infusion-related reactions	16 (19.8)	1 (1.2)	23 (29.5)	1 (1.3)

Abbreviations: N, number of participants in the safety population; n, number of participants within a subcategory. The response criteria used was RECIST 1.1. Data cut-off date = 5 January 2021.

**Table 5 cancers-16-01323-t005:** Phase 2: treatment-emergent adverse events related to study treatment occurring in ≥5% of drug-treated participants.

Parameter	Olaratumab Arm		Placebo Arm	
	Any Grade n (%)	Grade 3/4/5 n (%)	Any Grade n (%)	Grade 3/4/5 n (%)
Participants with ≥1 TEAE related to study treatment	75 (92.6)	62 (76.5)	77 (98.7)	60 (76.9)
Anemia	39 (48.1)	16 (19.8)	39 (50.0)	15 (19.2)
Fatigue	35 (43.2)	4 (4.9)	36 (46.2)	5 (6.4)
Diarrhea	31 (38.3)	4 (4.9)	26 (33.3)	4 (5.1)
Neutrophil count decreased	29 (35.8)	20 (24.7)	24 (30.8)	22 (28.2)
Platelet count decreased	26 (32.1)	6 (7.4)	28 (35.9)	9 (11.5)
White blood cell count decreased	23 (28.4)	13 (16.0)	15 (19.2)	12 (15.4)
Neutropenia	18 (22.2)	13 (16.0)	16 (20.5)	15 (19.2)
Infusion-related reactions	16 (19.8)	1 (1.2)	23 (29.5)	1 (1.3)
Peripheral sensory neuropathy	13 (16.0)	4 (4.9)	12 (15.4)	4 (5.1)
Thrombocytopenia	6 (7.4)	3 (3.7)	11 (14.1)	6 (7.7)

Abbreviations: N, number of participants in safety population; n, number of participants within a subcategory; TEAE, treatment-emergent adverse event. The response criteria used was RECIST 1.1. Data cut-off date = 5 January 2021.

## Data Availability

Data are contained within the article or Appendix A.

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
