# Peer review of "Results of a Randomized, Double-Blind, Placebo-Controlled, Phase 1b/2 Trial of Nabpaclitaxel + Gemcitabine ± Olaratumab in Treatment-Naïve Participants with Metastatic Pancreatic Cancer"

_cancers, 2024, doi:10.3390/cancers16071323_

Round 1

Reviewer 1 Report

Comments and Suggestions for Authors

·      As the authors acknowledged that pancreatic cancer is heterogeneous, it may achieve a better outcome to apply the treatment of anti-PDGFRα and chemotherapy to a selected cohort having higher expression of PDGFRα.

·      Figure 1, Panel A, “cohort 1 tolerated” and “cohort 1 not tolerated” appeared to be mixed up. Please check and revise if it’s a mistake.

Reviewer 2 Report

Comments and Suggestions for Authors

In this study, the authors evaluated the clinical efficacy of olaratumab , nabpaclitaxel, and gemcitabine in patients with pancreatic ductal adenocarcinoma in a randomized, double-blind, placebo-controlled trial. The study confirmed that the combination of olaratumab and chemotherapy did not significantly improve clinical outcomes in patients with metastatic PDAC.This study has scientific analytical methods and clear conclusions that could bring more attention to readers who are interested in PDAC treatment regimens.

However, I have some minor questions about it.

1. There does not seem to be a detailed description of the statistical methods?

2. The references of this manuscript are too few, the number of references can be increased appropriately in the discussion.

Reviewer 3 Report

Comments and Suggestions for Authors

This manuscript is well written, the data are presented well, and all conclusions are supported by the data.

Author Response

Dear Reviewer,

Thank you for your feedback on the manuscript.

Reviewer 4 Report

Comments and Suggestions for Authors

I am grateful for the opportunity to review the paper “Results of a randomized, double-blind, placebo-controlled, phase 2 trial of nabpaclitaxel + gemcitabine ± olaratumab in 3 treatment-naïve participants with metastatic pancreatic cancer". Your study focuses on determining the olaratumab dose and comparing overall survival (OS) in olaratumab arm vs placebo arms. This research addresses a significant clinical concern. I would like to congratulate the authors’ effort. I do have a few suggestions and concerns.

Minor points

# Title: I feel that it would be better to mention about Phase 1b trial.

# Correct me if I am wrong, but I feel figure 2 seems to contain incorrect description. For example, “on treatment” and “off treatment” appears opposite to me. In addition, the box of “Reason for discontinuation” looks the same in the intervention and control groups. Please check.

# Is it possible to include p-value in Table 1. Also, a history of pancreatectomy might be interesting to add to the analyses.

Round 2

Reviewer 1 Report

Comments and Suggestions for Authors

The authors' response is much appreciated.